# The Importance of Complement-Mediated Immune Signaling in Alzheimer’s Disease Pathogenesis

**DOI:** 10.3390/ijms25020817

**Published:** 2024-01-09

**Authors:** André F. Batista, Khyrul A. Khan, Maria-Tzousi Papavergi, Cynthia A. Lemere

**Affiliations:** 1Ann Romney Center for Neurologic Diseases, Brigham and Women’s Hospital, Harvard Medical School, Boston, MA 02115, USA; andrefelbatista@gmail.com (A.F.B.); khankhyrul1996@gmail.com (K.A.K.); mtzousipapavergi@bwh.harvard.edu (M.-T.P.); 2School for Mental Health and Neuroscience (MHeNs), Department of Psychiatry and Neuropsychology, Maastricht University, P.O. Box 616, 6200 MD Maastricht, The Netherlands

**Keywords:** complement system, neurodegeneration, Alzheimer’s disease, neuroinflammation, aging, brain development

## Abstract

As an essential component of our innate immune system, the complement system is responsible for our defense against pathogens. The complement cascade has complex roles in the central nervous system (CNS), most of what we know about it stems from its role in brain development. However, in recent years, numerous reports have implicated the classical complement cascade in both brain development and decline. More specifically, complement dysfunction has been implicated in neurodegenerative disorders, such as Alzheimer’s disease (AD), which is the most common form of dementia. Synapse loss is one of the main pathological hallmarks of AD and correlates with memory impairment. Throughout the course of AD progression, synapses are tagged with complement proteins and are consequently removed by microglia that express complement receptors. Notably, astrocytes are also capable of secreting signals that induce the expression of complement proteins in the CNS. Both astrocytes and microglia are implicated in neuroinflammation, another hallmark of AD pathogenesis. In this review, we provide an overview of previously known and newly established roles for the complement cascade in the CNS and we explore how complement interactions with microglia, astrocytes, and other risk factors such as TREM2 and ApoE4 modulate the processes of neurodegeneration in both amyloid and tau models of AD.

## 1. Introduction

Alzheimer’s disease (AD) is the most common form of dementia in the world and is characterized by a progressive loss of memory and other cognitive functions such as thinking and reasoning [1,2,3]. Currently, more than 55 million people worldwide are living with AD or related dementia, and this number is estimated to double every 20 years [2,4]. A better understanding of the disease mechanisms and development of efficient therapeutics are needed—both to combat the rising cost of healthcare associated with managing the disease and to address the trend towards a growing worldwide aging population which poses a risk factor for AD.

AD is characterized by two molecular pathological hallmarks: extracellular amyloid-β (Aβ) plaques and intracellular neurofibrillary tangles (NFTs) [1]. The aggregation of Aβ plaques and NFTs is associated with significant neurodegeneration and synaptic loss as well as neuroinflammation [1]. Notably, synaptic loss is the strongest correlate for clinical dementia and more specifically for memory impairment in AD [5,6].

The literature in the AD field has historically described observations of altered neuroimmune function using vague terms such as neuroinflammation or glial “activation”, whereas several studies have now confirmed that alterations in immune cell phenotypes, gene expression, and morphology are complex processes with wide-ranging implications in human disease [7,8,9]. Interestingly, genome-wide association studies (GWAS) implicate glial cells in several neurodegenerative disorders, including AD [10,11,12]. The “traditional” concept of neuroinflammation has been modified in recent years, as a remarkable cellular heterogeneity is being noticed through the molecular profiling of the central nervous system (CNS) cell population and single-cell analysis [7,8,11,13].

The complement system is a central component of innate immunity that serves as a first line of defense against pathogens, eliminating them and removing both apoptotic cells and debris [14,15]. In the brain, the complement system serves a crucial role during brain development [16]. Within this context, separate proteins and pathways of complement have been described as key players in the formation, development, migration, and refinement of neurons [17,18,19]. In addition, there is a growing body of work, in both animal models and humans, suggesting that aberrant complement regulation may underlie several neurodegenerative diseases [20,21,22,23,24,25,26]. In this review, we provide an overview of the data supporting the link between the complement system and the pathogenesis of AD and discuss new findings in this field.

## 2. Neuroinflammation and AD

Neuroinflammation has been described as a prominent feature in AD [27]. Since the 1980s, there have been reports of immune-related proteins and cells (microgliosis and astrogliosis) located close to and surrounding amyloid plaques [28,29,30,31,32,33,34,35]. Some evidence indicates that excessive brain inflammation is implicated in the pathogenesis of a number of neurological disorders [36], including AD, where increased levels of inflammatory mediators such as cytokines and chemokines are elevated in patients with the disorder [37,38]. In addition to inflammatory changes, progressive neuronal loss is observed, ultimately resulting in cognitive impairment [39].

Microglia and astrocytes have emerged as central players in both brain health and disease. Microglia are mononuclear phagocytes from the myeloid cell lineage and serve as the resident macrophages of the brain [40]. They originate from the yolk sac and migrate into the CNS during early embryonic development [41]. However, microglia are not evenly distributed throughout all brain regions [42]. In the normal physiological state, microglia exhibit a branched morphology with processes that lengthen and recede, allowing them to actively survey their immediate environment [43]. They also possess phagocytic capability and secrete several compounds, including trophic factors, cytokines, chemokines, nitric oxide, and reactive oxygen species that are responsible for immune defense and tissue maintenance [44]. In recent years, several seminal studies have recognized microglia as being pivotal during development, aging, and neurological diseases [7,8,16,17,18,20,22,23,24,45,46,47]. Remarkably, microglia can lose their homeostatic molecular signature and function during AD, and thus become inflammatory and contribute to neurodegeneration [47,48]. Moreover, genetic studies have indicated that most of the risk alleles for AD are preferentially or selectively expressed by microglia in the brain [49,50]. 

During development, microglia are heavily involved in a process known as synaptic pruning, in which they refine immature neuronal circuits by selectively enveloping and eradicating synaptic structures [16,17,18,51]. Studies have shown that the interruption of microglial-mediated synaptic pruning leads to defects in synaptic development and wiring [52,53,54]. Strong evidence from several studies also suggests that microglia are regulators of synaptic plasticity [55], a fundamental cellular mechanism important for learning and memory [56]. Conversely, in the AD brain, microglial function can be modified, for example by inflammasome signaling, in turn promoting neurotoxicity through the release of excessive inflammatory cytokines and contributing to excessive synapse loss by way of microglial-mediated synaptic elimination [22,48,57]. 

Astrocytes constitute one of the largest glial cell populations in the CNS and play several roles that are vital for optimal brain function including maintenance of ionic and osmotic concentration gradients, the control of neurotransmitter release, and the uptake and production of trophic factors that promote neuronal survival [58]. Furthermore, they secrete pro-inflammatory and/or anti-inflammatory cytokines in response to specific stimuli and are recognized as key modulators of neuroinflammation in the CNS [58,59,60]. Additionally, it has been shown that astrocytes play a pivotal role in synapse formation and function [61]. Thus, it is reasonable to infer that deficits in this cell type negatively impact brain health and trigger neurotoxicity in neurodegenerative diseases as shown by Liddelow et al. [59]. In particular, previous studies have established that astrocytes play a central role in the pathogenesis of AD [58,59]. Recently it was suggested that astrocytic reactivity may act as an upstream link between Aβ and tau pathology in early AD [62]. 

Along with microglia, astrocytes are also recognized as central regulators of the complement system in AD and, therefore, they can be expected to contribute to brain dysfunction via complement-mediated pathways [45,63]. The elevated expression and activation of various complement proteins have been observed in postmortem human AD tissue as well as several AD-like mouse models [22,64,65,66,67,68,69]. Furthermore, seminal studies have demonstrated that complement system signaling can contribute to AD pathogenesis [22,23]. Next, we provide a simplified overview of the complement system and how it mediates certain pathological aspects of AD. In addition, we discuss the relevant contributions from microglia and astrocytes to AD neuropathogenesis and neurodegeneration.

## 3. Complement System

As elucidated by Nonaka et al. [70], the complement cascade is an evolutionarily conserved system dating back more than a billion years. In 1895, complement was described as a heat-labile bactericidal component in serum [71]. Years later, one by one, the role of each essential component of the complement cascade was identified. The role of the complement system in innate immunity was discovered and elaborated on much later [72,73,74,75,76].

Currently, there are three well-defined complement-activation pathways: the classical, lectin, and alternative complement pathways (Figure 1). Notably, these pathways converge at C3 and downstream elements, which contribute to the formation of opsonins, anaphylatoxins, chemo-attractants, and the membrane attack complex (MAC) [77,78]. The complement system is regulated by both soluble proteins and cell surface proteins, which continually activate and inactivate these pathways to maintain homeostasis. When an immune reaction occurs against a pathogen, this system is amplified and mediates an inflammatory response that aids immune cells in fighting infection by enhancing (“complementing”) the ability of antibodies and phagocytic cells to remove microbes and damaged cells by targeting their plasma membrane.

Most of the complement proteins are produced in the liver; however, it is progressively evident that complement proteins, as well as their receptors and regulators, are expressed throughout the CNS [79]. Although the functions of the complement cascade in the periphery are well deciphered, its functions in the CNS are less clear and under current investigation.

Several reports have shown that complement plays a critical role in the developmental processes of synaptic function and pruning [16,17,18,80,81] (Figure 2). In this process, inactive synapses are cleared to allow bolstering and maturation of more robust connections [17,18]. During the refinement of the visual system, for example, early components of the classical complement pathway localize to synapses, tagging them for removal by microglia, which express complement receptors, e.g., CR3/CD11b [16,17,18,51]. When this process occurs inadequately, brain connectivity is affected due to inappropriate connections. Interestingly, changes in brain connectivity have been described in several neurological disorders, including traumatic brain injury [25], multiple sclerosis [12,82,83], stroke, and AD [84,85]. The past decade studies have also demonstrated a role for complement in neurodevelopmental disorders such as autism and schizophrenia [86,87,88]. Significant findings have also revealed that complement components C3a, C3a receptor (C3aR), and C3 are critical in adult hippocampal neurogenesis [89,90]. Further, the role of complement in the pathogenesis of the Zika virus in the CNS has been shown in mouse models [91]. Recent reports have also highlighted the key role of complement activation in the development of neurological manifestations following SARS-CoV-2 infection [92,93]. Importantly, studies in both animal models and human brain tissue indicate that the impairment of complement regulation may contribute to AD pathogenesis [22,23,26,33,63,65,66,67].

## 4. Complement System in the Pathogenesis of AD

In the last 20 years, complement activation has been broadly studied in animal models in the context of mechanisms related to AD. Most AD mouse models express multiple AD-associated gene mutations and thereby emulate features of early-onset forms of AD. Some of these AD mouse models overexpress various human mutant *APP* genes that cause early onset, autosomal-dominant, familial AD either alone or in combination with one or more *Presenilin-1* (*PSEN1*) autosomal dominant mutations that also lead to early onset familial AD. Newer knockin mouse models express physiological levels of mutant human *APP*. Overviews of these models are available on www.alzforum.org and in the following review [94]. Complement expression and its activation have been detected in AD brain tissue and mouse models [33,65,66,68,95,96]. It is now becoming increasingly clear that it serves a major role in AD pathology [16,21,23,97,98] (Figure 3). It has been shown that the classical complement proteins C1q, C3, and C4 co-localize with Aβ plaques and NFTs in memory-related areas in the AD brain [33,65,66,97,99]. Complement components and corresponding mRNA levels are also elevated in the AD brain and cerebrospinal fluid (CSF) [100,101]. Furthermore, increased levels of complement proteins have been reported in astrocyte-derived exosomes isolated from the plasma of AD patients [102]. A microarray study of young, healthy age, and AD brains demonstrated changes in complement-related gene expression in AD compared to age-matched controls [103]. GWAS have shown complement-related genes linked to AD [50,104,105]. Variants in the membrane protein complement receptor 1 (*CR1*) and the plasma regulator clusterin (*CLU*) genes of the complement pathway trigger late-onset AD, which is responsible for 95% of AD cases [104,106,107,108]. However, the mechanisms by which these complement genes affect AD pathology are still under investigation. Moreover, the role of complement in the inflammatory response observed in AD has also been widely explored in various reports [22,23,45,63,109,110,111].

C1q, the molecule that initiates the classical complement cascade, and C3, a downstream product and an “eat me” signal that attracts macrophages expressing C3 receptors (CR3), have been identified as key players in aging and AD pathogenesis [22,23,24,26,45,63,96,109,112]. Notably, fibrillar Aβ attaches to and activates C1q when immunoglobulins are not present, indicating that the classical complement cascade can be mobilized in AD independently of antibody-antigen immune complexes [28]. In the PS/APP mouse model, C1q co-localizes with fibrillar Aβ plaques [69]. In human AD postmortem brain tissue elevated C1q expression levels seem to positively correlate with Aβ plaques [113]. C1q has been demonstrated to localize with synapses and induce synaptic loss in AD mouse models [22] and has been suggested to act as an AD-specific modulator of the cellular crosstalk between microglia and astrocytes [114]. Recently, C1q was shown to be elevated in CSF-derived extracellular vesicles in AD patients compared to non-demented controls [115]. Furthermore, C1q has been shown to be increased in human and mouse brains with age [68,116]. On the other hand, deficiency of *C1q* in Tg2576 mice was shown to attenuate gliosis and synaptic degeneration without influencing Aβ levels [117]. Interestingly, deficiency of *C1q* in the 3xTg mouse model increased neurodegeneration [118]. These results suggest a critical role for the complement pathway in mediating AD pathology.

Recently, other complement components of synaptic dysfunction have been suggested to play a role in AD. Carpanini and colleagues showed that MAC fragments are located in the brains of mice in an APP^NL-G-F^ AD-like mouse model [119]. Concomitantly, treatment with an anti-C7 antibody reduced synapse loss in aged mice [119]. Furthermore, deficiency of terminal pathway (*C6*-deficient mice) rescued synapse loss [119]. These data suggest that other complement components also drive synapse dysfunction in AD and open new avenues of therapies for neurodegenerative diseases. 

## 5. Complement Component C3 and AD 

Complement C3 is the central molecule of the complement cascade, at which the different complement pathways merge (Figure 1) and serves a pivotal role in the immune system [77,78]. C3 has also been identified as a key player during brain development and aging [17,18,24,103]. More specifically, C3 levels have been shown to increase during aging and in AD patient brains and CSF, and overactivation of C3 has been associated with neuronal damage [26]. Following initiation of the classical complement cascade by C1q, C3 is activated leading to the elimination of pathogens [77,78]. Thus, knockout or inhibition of C3, which will inhibit all downstream components of complement activation, may serve as a promising therapeutic target in various disorders, including AD. However, blocking C3 throughout the body may reduce the protection against antigens. Therefore, a site-directed approach to directly lower C3 within the CNS would be more desirable.

We have been investigating the role of complement proteins during aging and AD pathogenesis for the last 20 years [22,23,24,67,69,96,112]. As mentioned above, synapse loss has been described as an early hallmark during aging and it correlates strongly with cognitive impairment in AD [5,6]. Reports have demonstrated that complement pathways mediate neurodegeneration during aging and AD [22,23,24,26]. In our work, we have shown an increase in C3 deposition at the CA3 region of the hippocampus in young adult mice (wild-type mice) 4 months in age, followed by a reduction in synaptic density and neurons in the same region in mice 16 months in age [24]. This suggests that the decrease in number of synapses in CA3 of the hippocampus is functionally significant during aging. We then determined whether C3 could contribute to age-dependent synapse loss in mice. We demonstrated in *C3* knockout mice (*C3KO*) that lifelong *C3* deficiency rescued synaptic degeneration during aging and protected against long-term potentiation (LTP) impairment in aged mice, improving their memory. Interestingly, *C3* deficiency also rescued neuronal loss at the CA3 region of the hippocampus in aged wild-type (WT) mice. Others have shown that *C3* deficiency resulted in elevated neurogenesis in the hippocampus of adult mice due to higher survival of newborn neurons [90]. These data reveal that there are underlying mechanisms by which the hippocampus becomes more susceptible to neurodegeneration during aging and point towards C3 as a viable target for studying brain aging and memory [24].

Next, our group evaluated the role of C3 in memory impairment during neuropathological changes in an amyloidosis mouse model, specifically in APPswe/PS1Δ9 mice [23]. First, we showed that lifelong *C3* deficiency in APPswe/PS1Δ9 mice rescued cognition and learning at 16 months of age. We had previously reported that aged *C3*-deficient J20 hAPP transgenic mice had a higher Aβ deposition and insoluble Aβ42 level in the hippocampus, whereas soluble Aβ levels were decreased [96]. These results indicate that inhibition of C3, and in effect its downstream activation products, might culminate in an age-dependent elevation in the aggregation of Aβ plaques and insoluble Aβ in the brain of these transgenic mice due to reduced phagocytic activity of microglia. 

In addition to the impact of C3 on Aβ burden, *C3* deficiency in APP/PS1 mice alleviated the microgliosis and astrogliosis in the hippocampus of the AD-like mice. The position and morphology of glial cells within and around plaques were altered in these mice, implying that *C3* deficiency may impact the glial cell response to plaques to favor the downregulation of some pro-inflammatory cytokine levels, including tumor necrosis factor-alpha (TNF-α), interferon-gamma (IFN-γ), and interleukin-12 (IL-12). Further, *C3* deficiency had protective effects against synaptic and neuronal loss in the hippocampal CA3 region in APP/PS1 mice [23] but not in J20 mice, in which gliosis was not altered and more neuritic dystrophy was observed. The results in our *C3*-deficient APP/PS1 mice were corroborated by Wu et al. [26] in another amyloid mouse model, suggesting that there is something unusual about the immune response in the J20 *C3*-deficient line. These results indicate that during neuroinflammation C3 may play a critical role that facilitates synaptic impairment.

## 6. Driving Mechanisms of Complement-Mediated CNS Dysfunction

A study by Xin et al. demonstrated through in vivo lipopolysaccharides (LPS) injections how the innate immune system drives synapse loss and memory impairment in mice [120]. In humans and mice, the endotoxin in LPS induces immune cells to produce pro-inflammatory cytokines that trigger inflammation. In the study, complement protein levels (C1q and C3) were increased in the hippocampi of the LPS-injected mice and this was associated with both phagocytic activity of microglia and memory loss [120]. This is consistent with the notion that peripheral inflammation, in this case caused by LPS, induces complement-mediated synapse loss and learning impairment in mice.

Nevertheless, the mechanisms by which the complement cascade regulates and affects neuronal function and dysfunction in the brain are relatively unknown. Some reports have proposed potential mechanisms by which complement may mediate neurodegenerative pathology. It is known that C1q, along with TNF-α and interleukin-1α (IL-1α), are secreted by activated microglia to stimulate the conversion of astrocytes into their reactive pattern (A1 astrocytes), contributing to neurodegeneration, synaptic loss, and impairment of synaptic pruning [59]. A1 astrocytes, which express elevated levels of C3, are found in several neurodegenerative diseases, including AD [59] (Figure 3). Complement C3 has been proposed as an astroglial initiator of nuclear factor-κΒ (NFκB) signaling through neuronal C3aR, which is a G-protein-coupled receptor expressed in many cell types [59,109]. NFκB/C3/C3aR signaling causes dendritic and synaptic loss in neurons, and hence, it makes sense that C3aR antagonist-treated APP/PS1 mice exhibit improved cognition [109]. 

A study by Propson et al. investigated the role of C3a/C3aR signaling in the context of blood–brain barrier (BBB) permeability during aging [121]. They showed that the C3a/C3aR pathway, specifically in endothelial cells of the brain, altered vascularity, increased BBB permeability, and enhanced microglial reactivity in aged WT mice. The deleterious effects of the signaling pathway were rescued by germline knock-out of C3aR or through elimination of C3aR using a pharmacological inhibitor [121]. The authors further elucidated the BBB vasculature phenotype which was exacerbated in AD-like PS19 tau-transgenic mice. Notably, C3aR ablation in this tau mouse model significantly improved vascular dysfunction. Taken together, these data suggest that the C3/C3aR signaling pathway is an important determinant of BBB integrity during aging and that this pathway contributes to vascular dysfunction in tauopathy [121]. Moreover, a role for C3/C3aR signaling in mediating the interaction between astrocytes, microglia, and neurons upon the presence of gut *Helicobacter pylori*-derived outer membrane vesicles can contribute to AD pathology [111]. These recent data reveal that *Helicobacter pylori* has a damaging effect on the brain and increases Aβ pathology via C3/C3aR signaling [111]. 

Microglia, the resident immune cells in the brain, are also known to express C3aR [122]. Lian et al. reported that C3 released from astrocytes interacts with C3aR on microglia and mediates Aβ pathology and neuroinflammation in AD, and these pathological features were ameliorated following treatment with a C3aR antagonist [45]. Further, *C3aR* deficiency rescued the dysregulated lipid profiles and ameliorated Aβ pathology and cognition improvement in an AD-like mouse model [123]. The authors also described metabolic-related changes in the transcriptome and suggested that targeting this pathway (C3a) could be a therapeutic strategy for AD [123]. C3 and CR3 also contribute to Aβ clearance mediated by microglia, thereby supporting the beneficial functions of microglia during AD pathogenesis [96,112,124,125]. Furthermore, the microglial CR3 can regulate Aβ homeostasis via proteolytic activity, independent of phagocytosis [126].

## 7. Complement Components as Therapeutical Targets in AD

Therapeutic inhibition of the complement system has been highlighted as a promising therapeutic target for the treatment of neurodegenerative diseases [20]. In addition to the C3aR antagonist mentioned above, other complement proteins have been identified as potential targets in AD immunotherapy. For instance, inhibition of the complement C5a receptor (C5aR1) with an antagonist (PMX205) reduced Aβ pathology and improved cognition in transgenic AD mice [127]. The antagonist reduced dystrophic neurites and Aβ levels and rescued the loss of pre-synaptic markers in Tg2576 mice [128]. It was also associated with AD rescue of memory loss and partially restored microglial homeostatic genes [128]. Another study has shown that EP67, a modified C5a receptor agonist in an AD amyloidosis mouse model (5xFAD model), boosted microglial phagocytosis of both fibrillar and non-fibrillar Aβ, improving cognitive impairment and protecting against synapse loss [129]. Furthermore, another group showed that overexpression of *C5a* caused a cognitive decline in mice and increased the levels of complement components (e.g., C3) in the Arctic model of AD [130]. Another study also showed that *C5a* deficiency reduced astrogliosis and microglial activation [128]. A 2020 study by Wang et al. demonstrated that overexpression of *CD55*, an inhibitor of complement pathways expressed in neurons in response to inflammation, led to decreased elimination of synapses in the dentate gyrus of mice hippocampi [131]. In addition, overexpression of *CD55* in the CA1 region of the mouse hippocampus also resulted in less dissociation of synapses and less forgetting of remote memories [131].

More recently, C8γ, one of the three subunits of C8 (α, β, γ), which constitutes a component of MAC, was identified for the first time as a neuroinflammation inhibitor in the AD brain [132]. Researchers revealed higher C8γ levels in the brain, CSF, and plasma of AD patients. In the mouse brain, C8γ was found to mainly localize in astrocytes, perhaps as an attempt to prevent complement overactivation. C8γ has been shown to inhibit glial hyperactivation, neuroinflammation, and cognitive decline in AD mouse models, highlighting its role as an immune modulator (acting as an anti-inflammatory marker). Thus, C8γ may have potential as a therapeutic target in AD and other neurological disorders [132].

## 8. Crosstalk between the Complement System and Other Inflammatory Players in AD

It is important to consider the interaction between complement proteins and cytokines since pro-inflammatory cytokines have long been implicated in neurodegenerative mechanisms. One such class of cytokines, Type I interferons (IFNs), are produced by the innate immune system especially in response to viral infections [133]. Additionally, IFNs are predominantly expressed by immune cells upon activation of intracellular innate immune sensors [133]. In AD, its role has been extensively investigated. One study demonstrated the role of IFN responses leading to neuroinflammation and synaptic loss in AD [46]. Overactivation of the IFN pathway was observed in amyloidosis mice models with its activation manifested in plaque-associated microglia. Furthermore, IFN activation led to microglial activation and pronounced synaptic degeneration in mice. Additionally, IFN stimulated the complement cascade, with an increase in complement proteins (e.g., C1q and C3) in both in vitro and in vivo models. Notably, APP^NL-G-F/NL-G-F^ mice that were treated with IFN-blocking antibodies showed protection of synapses and decreased C3 expression. Lack of IFN signaling also showed protective effects in APP/PS1 and 5xFAD mice, as reflected in decreased synapse loss and microglial activation [134,135]. In *C3*KO mice that received recombinant IFN, the brains were protected against synaptic degeneration, despite the activation of microglia. These findings suggest that the IFN–C3 axis is important in promoting synapse loss in AD-like mice models [46].

Apolipoprotein-E (ApoE) is another molecule that has been linked to AD, atherosclerosis, and other inflammatory conditions [136]. In particular, the *APOE4* allele has been associated with an elevated risk of AD [136,137]. In an extensive report in 2014, Yin et al. showed that the complement pathway in A*POE*-deficient mice is activated in atherosclerosis and AD [138]. Interestingly, this group has shown that ApoE binding to activated C1q inhibits the activation of the classical complement cascade. However, the formation of C1q-ApoE complexes was seen within Aβ plaques and arteries and correlated with cognitive impairment and atherosclerosis [138]. Taken together, these results suggest that ApoE may be a checkpoint inhibitor for activation of the classical complement cascade. The C1q–ApoE complex was suggested to further affect AD pathology by modulating microglia-mediated synaptic clearance [139]. Furthermore, upon intracerebroventricular injections of Aβ oligomers, a profound increase in C3 reactivity was observed in the choroid plexus [140]. Aβ oligomers are synaptotoxins linked to synapse dysfunction in AD [1]. Thus, it may be possible that the accumulation of cerebral Aβ oligomers and *C3* upregulation in the choroid plexus may contribute to the early stages of AD. On the other hand, the *APOE2* allele has been associated with a decreased risk of AD [141,142]. Panitch and colleagues revealed an interaction of ApoE2 with complement components C4A and C4B, as well as with HSPA2 oligodendrocyte-specific protein, which was significantly linked to the complement cascade, resulting in a neuroprotective effect against AD [143]. This finding indicates a strong link between the complement system and the protective effects of the *APOE2* allele in AD. 

The triggering receptor expressed on myeloid cells 2 (TREM2) is an innate immune receptor of the immunoglobulin superfamily, expressed in different populations of myeloid cells including microglia in the CNS [144]. Its variants have been associated with an increased risk for AD [145,146]. Functionally, TREM2 is required for important cellular functions such as chemotaxis, maintenance of energy metabolism, and phagocytosis of Aβ fibrils [147]. Along with ApoE, TREM2 is also upregulated in microglia responding to brain lesions [8,48,144]. Pointing towards a possible interaction between microglial activity and TREM2, it was demonstrated that in TREM2 knockout mice, there is persistent impairment of synaptic refinement during the early stages of brain development [53], suggesting that TREM2 regulates microglial activity at this critical time. Furthermore, TREM2 receptor deficiency results in impaired synaptic clearance, elevated dendritic spine density, and enhanced excitatory neurotransmission [148]. These findings indicate that microglial TREM2 could impact the process of synapse elimination. Importantly, studies have inferred an association between TREM2 and C1q in microglia-mediated synaptic pruning due to their co-localization around Aβ plaques [149,150]. As mentioned above, C1q also interacts with ApoE, forming the C1q–ApoE complex. Qin and colleagues recently highlighted as a potential underlying mechanism of the microglia-mediated synaptic loss induced in AD the interaction between TREM2, APOE, and C1q [139].

TYROBP (tyrosine motif-binding protein, aka DAP12 or DNAX-binding protein-12) regulates multiple genes involved in microglia phagocytosis, and its expression is increased in aged human AD brains [151] and AD-like mouse models [152]. Interestingly, TYROBP serves as an adaptor for some immune receptor ectodomains, including TREM2 and CR3 [153]. A study by Haure-Mirande et al. proposed that the absence of TYROBP in a mouse model of AD-type cerebral Aβ amyloidosis (APP/PSEN1 mouse model) protected against the dysregulation of complement transcriptomic network during AD pathogenesis [154]. Furthermore, the absence of TYROBP decreased C1q expression in APP/PSEN1 mice [154,155]. The absence of TYROBP in these AD-like mice led to decreased C3 levels in the brain in the early stage of Aβ pathology [153]. Together, these results demonstrate an important interaction between TYROBP and the complement pathway which should be further explored in the future.

## 9. Complement and Tau Pathology

Previous work has shown that amyloid pathology long precedes clinical AD symptoms, and that tau pathology and synapse loss correlate strongly with cognitive impairment [156,157]. Emerging evidence suggests that tau pathology is also associated with complement [26,158,159,160]. In fact, overexpression of a C3 inhibitor *(sCrry)* reduced the level of hyperphosphorylated tau in aged P301L/sCrry double-transgenic mice [158]. More recently, interactions between C3, tau, ApoE4, and Aβ have been identified. In a detailed study, Bonham et al. reported an interaction between *APOE4* genotype and CSF C3 levels on CSF levels of Aβ and pTau in AD patients in the ADNI study [161]. However, CSF C3 was only associated with CSF pTau if it was also associated with CSF Aβ [161]. Additionally, in a tau mouse model expressing human *APOE4*, microglial C3 gene expression was observed to be elevated [162].

Another study demonstrated that C1qa levels are enhanced on postsynaptic dendritic spines in the Tau P301S (PS19) mouse model and in human tissue, and a positive correlation was observed between C1qa and phospho-tau levels [159]. Interestingly, the inhibition of C1q by C1q-blocking antibodies rescued the synapse loss induced by microglia in these transgenic mice [159]. Another report has shown that C3 and C3aR are positively associated with cognitive decline in the brains of AD patients [160]. This same work also illustrated the role of C3/C3aR signaling in tau pathogenesis in PS19 mice deficient in *C3ar1* [160]. In this study, the authors identified a mechanism dependent on Stat3 by which C3aR regulates tau pathology in tau transgenic mice. Additionally, when the PS19 tau mice were crossed with mice deficient in *C3ar1*, there was an improvement in tau pathology, neuroinflammation, synaptic and cognitive impairment [160]. Additionally, Wu et al. indicated that loss of *C3* in PS19 mice ameliorated neuronal loss and neurodegeneration, while positively affecting neurophysiological and behavioral outcomes [26]. The researchers also revealed that brain and CSF C3 levels are increased in AD patients [26]. Recently, a study demonstrated a mechanism by which astrocytes and microglia engulf excitatory and inhibitory synapses, respectively, in the PS19 tau mouse model [163]. Remarkably, excitatory synapses were found in the lysosomes of astrocytes, whereas inhibitory synapses were found in the microglial lysosomes. *C1q* deficiency reduced synaptic pruning by both immune cell types and alleviated synapse loss in the PS19 mouse model. These findings support a role for complement-tau interaction in the pathogenesis of AD.

## 10. Future Directions

In recent years, sophisticated in vitro tools have been used to study the interaction between different players in the complement pathway as it relates to neurodegenerative diseases, including AD. In their 2021 paper, Guttikonda and colleagues unveiled a new triculture system that combines pure populations of neurons, astrocytes, and microglia [114]. They showed that under inflammatory conditions (triggered by LPS), microglia release factors that stimulate astrocytes to produce C3, which feeds back on microglia and triggers them to generate their own supply. The same inflammatory loop was engaged when neurons carried the *APP* Swedish familial Alzheimer’s mutation, with both microglia and astrocytes participating. Altogether, the data suggested that microglia containing C3 kick off an inflammatory loop by releasing TNF-α. This provokes astrocytes to release large amounts of C3, which stimulates microglia to produce even more of the complement protein. Mutant APP exacerbates this crosstalk, implicating this pathway in AD [114]. Although this triculture model presents an exciting new system to promptly define cellular pathways, it must be noted that findings from this study still need to be validated in vivo, because one limitation of such cell culture models is that they are unable to fully capture age- and time-dependent effects which are in looking at the big picture in AD pathology. In addition, inducible conditional knock mouse models of various components of the complement system are underway to globally or cell-specifically knockdown complement proteins after brain development or at any stage during AD pathogenesis. These models should be helpful in understanding when and where complement-targeted therapies might be most effective during the disease process. Moreover, because complement C1q binds antibody–antigen complexes and activates the classical complement cascade, it is possible that anti-amyloid antibodies that bind fibrillar amyloid may bind vascular amyloid deposits, become tagged by C1q, and initiate complement activation. This could potentially result in the opsonization/phagocytosis of the immune complexes (e.g., C3b/CR3), recruitment of immune cells to the local area, and secretion of pro-inflammatory molecules (C3a/C3aR and C5a/C5aR), and cell lysis by C5b-9 (MAC). As a result, BBB leakiness, edema, and, possibly microhemorrhages may occur, most of which have been observed in a subset of AD patients treated with clinical anti-amyloid antibodies. We are actively pursuing this research question. Lastly, complement provides the body with protection against pathogens, so the development of complement therapies to target complement within the CNS without disturbing it in the periphery warrants further investigation. Such treatments might be amenable to other neurodegenerative diseases in which complement appears to be involved such as Parkinson’s disease, Multiple Sclerosis, and Huntington’s disease to name a few.

## 11. Conclusions

The role of the complement system during both healthy and diseased brain states has been widely investigated. This review summarizes pertinent evidence that emphasizes the pivotal role of aberrant complement cascade activation in neurodegenerative diseases, especially in AD. We also show that different proteins in the complement pathway interact with other key players in neuroinflammation and neurodegeneration such as astrocytes, microglia, ApoE, and TREM2, among others. The findings discussed in this report support the notion that targeting complement proteins, including cell-specific targeting, may represent a promising new approach to protect the brain in AD and other neurodegenerative diseases, slow the progression of the disease, and potentially improve cognitive and memory outcomes for AD patients in the future. 

## Figures and Tables

**Figure 1 ijms-25-00817-f001:**
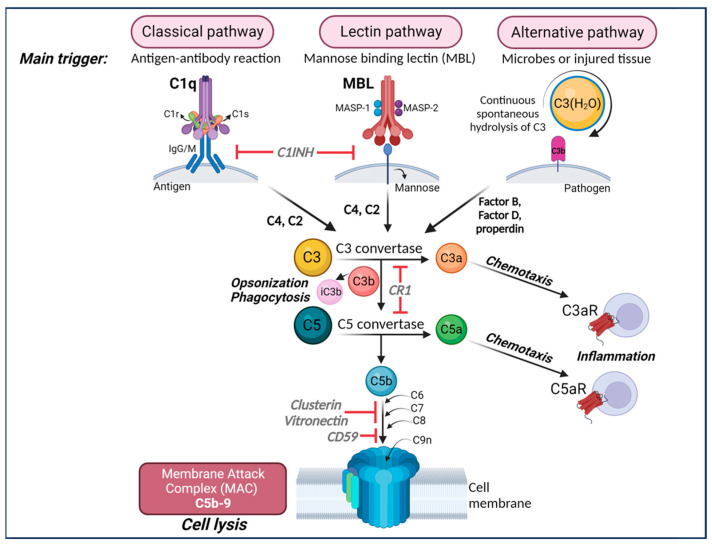
The complement cascade. The complement can be activated by three separate pathways: the classical, lectin, and alternative pathways. Cleavage of C3 is central to complement activation and its downstream effects lead to lysis and inflammatory signaling through the formation of the membrane attack complex (MAC). The classical pathway is activated upon binding of C1q to antigen–antibody complexes, consequently activating the two serine proteases C1r and C1s to form a complex with C1q known as the C1 complex. The C1 complex causes the cleavage of C4 and then C2, whose fragments combine to form C3 convertase (C4bC2a). The lectin pathway is initiated by binding carbohydrates such as mannose-binding protein (MBP) and ficolins on the surface of pathogens which activate two proteases, MASP1 and MASP2 (mannose-binding serine proteases 1 and 2), causing cleavage of C4 and then C2, whose fragments combine to form the same C3 convertase as in the classical pathway. The alternative pathway is always “on” at low levels in the blood and is further activated by pathogens including viruses, fungi, bacteria, LPS, etc., leading to the spontaneous hydrolysis of C3 and amplification of C3b, which binds Factor B and is cleaved by Factor D to form a different C3 convertase (C3bBb). Both C3 convertases cleave C3 into C3a and C3b. C3b can combine with the two different C3 convertases to form C5 convertase (C4bC2aC3b in the classical and lectin pathways and C3bBbC3b in the alternative pathway) which cleaves C5 into C5a and C5b. C3a and C5a are anaphylatoxins that bind their respective receptors, C3aR and C5aR, to recruit immune cells to sites of injury or infection which then secrete pro-inflammatory cytokines (Note: There are two forms of C5aR. C5aR1 is pro-inflammatory while C5aR2 is thought to be anti-inflammatory). Complement component C3b and its inactivated product, iC3b, opsonize or tag pathogens or immune complexes or weak synapses for phagocytic removal by microglia that bear their receptor, CR3. Complement component C5b binds C6, C7, C8, and finally multiple copies of C9 to form a channel (C5b-9 or MAC) in the cell membrane of a pathogen or cell, which induces lysis. Figure created with BioRender.com (accessed on 14 December 2023).

**Figure 2 ijms-25-00817-f002:**
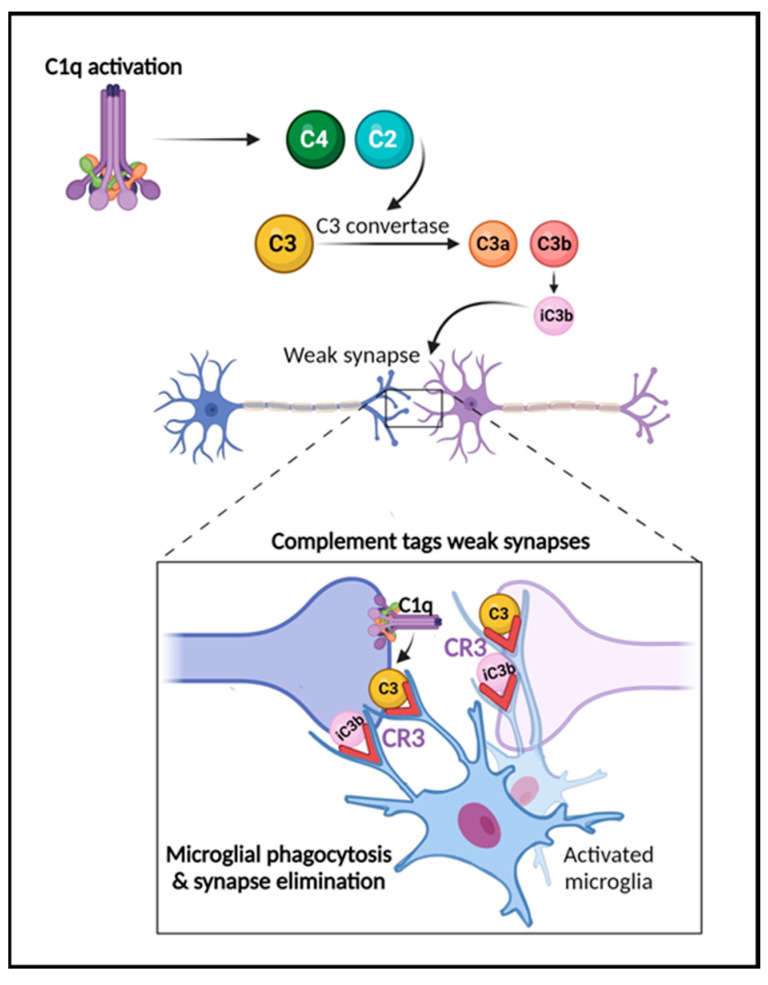
The role of the complement system in synapse elimination. The complement system serves a crucial role in normal brain development, aging, and AD progression by regulating synaptic pruning and elimination, as well as neuronal migration. The activation of the classical pathway leads to C1q expression causing cleavage of C4 and C2 and resulting in expression of C3 and iC3b. C3 and iC3b tag weak or unhealthy synapses representing an “eat me” signal. Microglia, which are responsible for maintaining homeostasis and surveilling the function of synapses, carry CR3 receptors (red) that recognize the tagged synapses and proceed with eliminating them via phagocytosis (synaptic engulfment). In aging and AD, this process has been linked to neurodegeneration. Figure created with BioRender.com (accessed on 14 December 2023).

**Figure 3 ijms-25-00817-f003:**
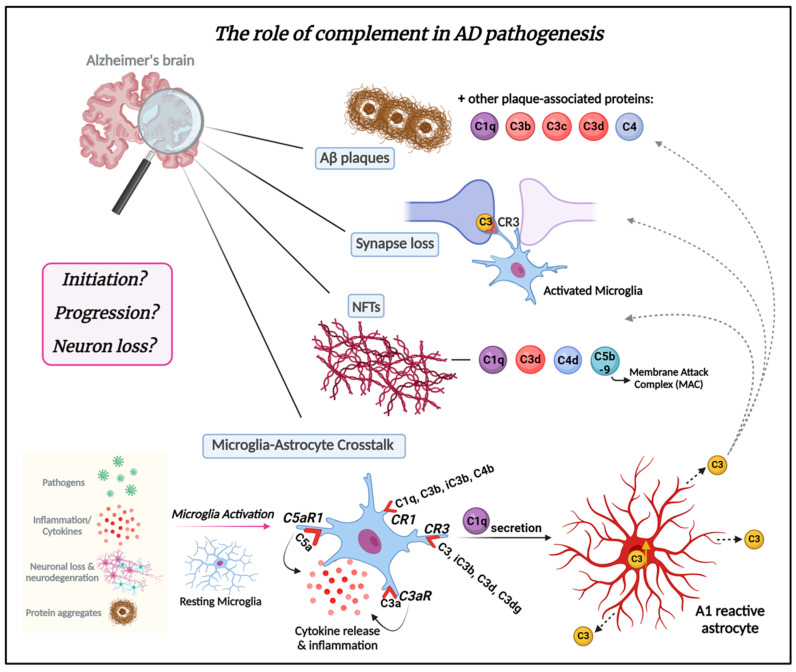
The role of the complement system in AD pathogenesis. The complement system plays a crucial role in AD pathogenesis. Complement proteins are deposited in amyloid plaques in Alzheimer’s brains and can be detected at early stages of plaque formation. Complement proteins C1q, C3b, C3c, and C3d are associated with amyloid plaques and dystrophic neurites therein throughout AD pathogenesis. Synapse loss is another critical feature of AD. Complement C3/C3b has emerged as a key player in synapse elimination in AD by tagging weak synapses and representing an “eat me” signal. Microglia carry CR3 receptors that recognize the tagged synapses and eliminate them via phagocytosis. Complement proteins, such as C1q, C3d, C4d, and C5b-9 also co-localize with neurofibrillary tangles (NFTs) in AD. The crosstalk between microglia and astrocytes is a significant part of AD pathogenesis and progression and may be mediated by complement. Microglia, the resident macrophages of the brain, can be activated in response to the presence of pathogens such as bacteria or viruses in the CNS, as well as by inflammatory signals (cytokines), signals released by injured or dying neurons, or in response to neurodegeneration. In addition, amyloid-β protein and tau aggregates can activate microglia. Once activated, microglia undergo morphological changes and release various molecules, including complement proteins such as C1q. Microglia express complement receptors CR1, CR3, C3aR, and C5aR1. CR1 binds complement opsonins C1q, C3b, iC3b, and C4b. CR3 binds C3 and its fragments C3d and C3dg, as well as iC3b (primary ligand). C3aR and C5aR1 (also known as CD88) specifically bind complement components C3a and C5a, respectively. The binding of these molecules to their corresponding receptors induces inflammation and cytokine release. C1q secreted by activated microglia induces A1-reactive astrocytes. A1 astrocytes are characterized by increased C3 expression and secretion. C3 released from A1-reactive astrocytes deposit on amyloid plaques, weakened synapses, and NFTs. It remains unknown whether complement is part of the initiation of AD pathology or drives the progression of the disease leading to neuron loss. Understanding the complex interplay between the complement system and AD is crucial for gaining insights into the molecular mechanisms driving AD and for the development of targeted therapeutic interventions. This knowledge not only holds the potential for addressing AD but also has broader implications for treating other neurodegenerative diseases. Figure created with BioRender.com (accessed on 4 January 2024).

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
