# Peer review of "The Importance of Complement-Mediated Immune Signaling in Alzheimer’s Disease Pathogenesis"

_ijms, 2024, doi:10.3390/ijms25020817_

Round 1
Reviewer 1 Report
Comments and Suggestions for Authors
Review on the manuscript of Batista, A. F. et al.: “The importance of complement-mediated immune signaling in Alzheimer’s disease pathogenesis”.
This manuscript represents a review on the roles of complement cascade in the CNS and how complement interactions with microglia, astrocytes, and other factors are involved in neurodegeneration, particularly in Alzheimer’s disease.
Overall, I felt that this topic is of great interest, as Alzheimer’s disease represent a neurodegenerative disease with high incidence and prevalence in the world population. Therefore, the elucidation of the disease mechanisms could help in developing new tools or therapies for disease control. The manuscript is very clear and, generally, well written. In addition, I consider that the manuscript is precise on the question that authors proposed to review. Thus, the issues that arise to me are listed below, so, I hope Authors find the following comments and suggestions useful.
1 – I recommend Authors to increase the size of Figure 1 to make easier the interpretations by the readers.
2 – In some parts of the manuscript, sentences are very long, making difficult to follow the idea. For example, the following sentence can be easily broken in 2 sentences - “Encoding the membrane protein complement receptor 1 (CR1) and encoding the plasma regulator clusterin (CLU) are genes of the complement pathway that trigger late-onset AD, responsible for 95% of AD cases [103, 105-107], however, the mechanisms by which these complement genes affect AD pathology are still under investigation”.
3 – In my opinion, the section of Future directions should provide research questions that could be interesting to explore in the near future. Based on this, I recommend Authors to go beyond the triculture model and indicate some ideas, including experiments or models, that could help elucidating the role of complement in CNS function and degeneration.
4 – The review is particularly focused in Alzheimer’s disease. However, it could be interesting to discuss in the sections of Future directions or Conclusions whether the findings shown for this pathology could be translated to other neurodegenerative diseases.
5 – I encourage Authors to make an additional figure summarizing the role of complement in Alzheimer’s disease.
Author Response
REVIEWER 1: Authors’ responses to the reviewer’s comments
Review on the manuscript of Batista, A. F. et al.: “The importance of complement-mediated immune signaling in Alzheimer’s disease pathogenesis”.
This manuscript represents a review on the roles of complement cascade in the CNS and how complement interactions with microglia, astrocytes, and other factors are involved in neurodegeneration, particularly in Alzheimer’s disease.
Overall, I felt that this topic is of great interest, as Alzheimer’s disease represent a neurodegenerative disease with high incidence and prevalence in the world population. Therefore, the elucidation of the disease mechanisms could help in developing new tools or therapies for disease control. The manuscript is very clear and, generally, well written. In addition, I consider that the manuscript is precise on the question that authors proposed to review. Thus, the issues that arise to me are listed below, so, I hope Authors find the following comments and suggestions useful.
Response: We greatly appreciate Reviewer 1’s positive impression of our review article and helpful feedback. We have made all of the suggested changes to the manuscript and believe this will enhance the readability and overall quality of the manuscript. Thank you!
1 – I recommend Authors to increase the size of Figure 1 to make easier the interpretations by the readers.
Response: We have increased the size of Figure 1.
2 – In some parts of the manuscript, sentences are very long, making difficult to follow the idea. For example, the following sentence can be easily broken in 2 sentences - “Encoding the membrane protein complement receptor 1 (CR1) and encoding the plasma regulator clusterin (CLU) are genes of the complement pathway that trigger late-onset AD, responsible for 95% of AD cases [103, 105-107], however, the mechanisms by which these complement genes affect AD pathology are still under investigation”.
Response: We apologize for the lengthy sentences and have shortened sentences or broken up long single sentences into 2 sentences in 12 places as highlighted in the revised manuscript.
3 – In my opinion, the section of Future directions should provide research questions that could be interesting to explore in the near future. Based on this, I recommend Authors to go beyond the triculture model and indicate some ideas, including experiments or models, that could help elucidating the role of complement in CNS function and degeneration.
Response: We are grateful for this opportunity to further speculate on interesting research avenues on the role of complement in AD, including novel mouse models, as well as potential therapeutic strategies to target CNS complement while leaving peripheral complement intact to protect patients with AD, and perhaps other neurodegenerative diseases, from infection.
4 – The review is particularly focused in Alzheimer’s disease. However, it could be interesting to discuss in the sections of Future directions or Conclusions whether the findings shown for this pathology could be translated to other neurodegenerative diseases.
Response: As described in the previous response, we have added to Discussion the possibility that complement targeted therapies for AD may be efficacious in other neurodegenerative diseases.
5 – I encourage Authors to make an additional figure summarizing the role of complement in Alzheimer’s disease.
Response; We appreciate this suggestion and have made Figure 3 to illustrate how complement may play a role in AD.
Reviewer 2 Report
Comments and Suggestions for Authors
This review is the overview of the recent researches about the roles of the complement system in the neurodegenerative diseases’ pathogenesis, specifically AD and sheds light on their treatments in the point of involvements of the complement system. It is theoretical, really interesting and might suggest the new possible approach to the treatments of the neurodegenerative diseases. I have just one concern if possible. There are lots of researches quoted and some of them use several AD-model animals such as PS/APP, Tg2576, APPNL-G-F AD-like, 3xTg, J20hAPP, J20 mice and so on which appear around on page 6-9. I think those various name of model animals are confused and might want to be explained a little more, for example the differences, to realize the context for readers.
Author Response
REVIEWER 2: Authors’ response to the reviewer’s comments
This review is the overview of the recent researches about the roles of the complement system in the neurodegenerative diseases’ pathogenesis, specifically AD and sheds light on their treatments in the point of involvements of the complement system. It is theoretical, really interesting and might suggest the new possible approach to the treatments of the neurodegenerative diseases. I have just one concern if possible. There are lots of researches quoted and some of them use several AD-model animals such as PS/APP, Tg2576, APPNL-G-F AD-like, 3xTg, J20hAPP, J20 mice and so on which appear around on page 6-9. I think those various name of model animals are confused and might want to be explained a little more, for example the differences, to realize the context for readers.
Response: We appreciate Reviewer 2’s overall positive comments and review of our manuscript. While it would take up too much space to describe all of the various AD mouse models in detail and distract from the overall message, we have included a short overall description summarizing the general categories of these animal models and provide a website (www.alzforum.org) and a new reference that provide much more detailed information regarding each of the models discussed here. We hope this will satisfy the reviewer as these resources are very helpful in understanding the critical components, temporal pathogenesis, phenotypes and utility and limitations of each animal model.